# Metabolically Improved Stem Cell Derived Hepatocyte-Like Cells Support HBV Life Cycle and Are a Promising Tool for HBV Studies and Antiviral Drug Screenings

**DOI:** 10.3390/biomedicines10020268

**Published:** 2022-01-26

**Authors:** Tine Tricot, Hendrik Jan Thibaut, Kayvan Abbasi, Ruben Boon, Nicky Helsen, Manoj Kumar, Johan Neyts, Catherine Verfaillie

**Affiliations:** 1Stem Cell Institute, Rega Institute KU Leuven, 3000 Leuven, Belgium; ruben.boon@kuleuven.be (R.B.); nicky.helsen@ismar.com (N.H.); manoj.kumar@kuleuven.be (M.K.); 2Department of Microbiology, Immunology and Transplantation, Virology and Chemotherapy, Rega Institute KU Leuven, 3000 Leuven, Belgium; kayvan.abbasi@kuleuven.be (K.A.); johan.neyts@kuleuven.be (J.N.); 3Department of Microbiology, Immunology and Transplantation, Translational Platform Virology and Chemotherapy (TPVC), Rega Institute KU Leuven, 3000 Leuven, Belgium; 4Laboratory for Functional Epigenetics, Department of Human Genetics, Rega Institute KU Leuven, 3000 Leuven, Belgium; 5Ismar Healthcare NV, 2500 Lier, Belgium

**Keywords:** hepatitis B virus, stem cell differentiation, hepatocytes

## Abstract

More than 300 million people worldwide are diagnosed with a chronic hepatitis B virus (HBV) infection. Nucleos(t)ide viral polymerase inhibitors are available on the market and can efficiently treat patients with chronic HBV. However, life-long treatment is needed as covalently closed circular DNA (cccDNA) persists in the hepatocyte nucleus. Hence, there is a high demand for novel therapeutics that can eliminate cccDNA from the hepatocyte nucleus and cure chronically infected HBV patients. The gold standard for in vitro HBV studies is primary human hepatocytes (PHHs). However, alternatives are needed due to donor organ shortage and high batch-to-batch variability. Therefore, human pluripotent stem cell (hPSC)-derived hepatocyte-like cells (HLCs) are being explored as an in vitro HBV infection model. We recently generated hPSC lines that overexpress three transcription factors (HC3x) and that, upon differentiation in a high amino-acid supplemented maturation medium, generate a more mature hepatocyte progeny (HC3x-AA-HLCs). Here, we demonstrate that HBV can efficiently infect these HC3x-AA-HLCs, as was shown by the presence of HBV core (HBc) and surface antigens. A clear increasing release of HBV surface and e antigens was detected, indicating the formation of functional cccDNA. Moreover, back-titration of culture supernatant of HBV-infected HC3x-AA-HLCs on HepG2-NTCP cells revealed the production of novel infectious HBV particles. Additionally, an increasing number of HBc-positive HC3x-AA-HLCs over time suggests viral spreading is occurring. Finally, the HC3x-AA-HLC model was validated for use in antiviral drug studies using the nucleoside reverse-transcriptase inhibitor, lamivudine, and the HBV entry inhibitor, Myrcludex B.

## 1. Introduction

Hepatitis B virus (HBV) is a non-cytopathic virus that belongs to the family *Hepadnaviridae* and is transmitted via blood or sexual contact [1]. Although prophylactic vaccines are available on the market, almost 300 million people have a chronic HBV infection and are at a high risk of developing liver fibrosis; liver cirrhosis; and, eventually, hepatocellular carcinoma (HCC) [2,3]. Currently, two types of approved therapies for HBV exist: (pegylated) interferon α (IFNα) and nucleoside/nucleotide reverse-transcriptase inhibitors (NRTIs). NRTIs, such as lamivudine, entecavir, and tenofovir, inhibit viral replication; however, patients need life-long treatment as the covalently closed circular DNA (cccDNA) is retained in the hepatocyte nucleus. Treatment with pegylated IFNα is effective only in a subgroup of patients and may cause severe side effects [4]. Due to these limitations, there is still a high need for novel anti-HBV antivirals that can cure rather than just control HBV infection in chronically infected patients.

Primary human hepatocytes (PHHs) are the gold standard to study HBV in vitro and screen for potential anti-HBV antivirals. Unfortunately, the major drawback of in vitro cultured PHHs is their limited availability and the fact that they rapidly lose functionality upon culturing, reflected by the decreased activity of most drug metabolizing enzymes, rendering them suboptimal for drug screenings. In addition, not all batches of PHHs can be readily infected with HBV [5,6,7]. Hepatoma cell lines, HepaRG and HepG2 have been used as alternative in vitro HBV models. However, these cell lines have their drawbacks as they are transformed cells and, in case of the HepG2 cell line, require overexpression of the HBV entry receptor, sodium taurocholate co-transporting polypeptide (NTCP) [8,9]. Moreover, these in vitro models do not support substantial viral spreading over time, a phenomenon that happens efficiently in vivo, and only a modest amount of cccDNA is detectable in these cell lines upon HBV infection [10].

To overcome the aforementioned limitations, human-pluripotent-stem-cell-derived hepatocyte-like cells (hPSC-HLCs) are being investigated as an in vitro model for HBV. Previously, several research groups demonstrated that hPSC-HLCs could support the infection with hepatotropic viruses, including hepatitis B virus (HBV) [7,11,12,13], hepatitis C virus (HCV) [14,15], hepatitis E virus (HEV) [16] and dengue virus (DENV) [17]. Even though hPSC-HLCs resemble PHHs by their expression of various essential hepatocyte markers, including hepatocyte nuclear factor (HNF) 4α, albumin (ALB), *NTCP* and α-1 antitrypsin (AAT), they are immature, with high expression of the hepatoblast marker, α-fetoprotein (AFP), and low expression of mature hepatocyte markers, such as HNF6 and drug metabolizing enzyme cytochrome P450 (CYP) 3A4, as compared to adult PHHs [18,19,20].

Recently, we demonstrated that the inducible overexpression of three transcription factors (TFs)—*hepatocyte nuclear factor*
*(HNF)1α*, *prospero homeobox protein 1*
*(PROX1)* and *forkhead box protein A3*
*(FOXA3)* (termed HC3x cells)—combined with the use of an optimized medium containing high concentrations of an amino acid mix with additional 2% glycine (HC3x-AAGly-HLCs), significantly increased mitochondrial activity of HLCs [21,22]. Aside from significantly improving cellular metabolism, the TFs and optimized culture medium induced global hepatocyte maturation and functionality. HC3x-AAGly-HLCs show significant CYP450 and can correctly classify hepatotoxic and non-toxic drugs significantly better than HepG2 or HepaRG cells [22,23]. Of note, when HC3x-hPSCs were cultured in medium only supplemented with high concentrations of amino acids (AA) but not extra glycine (HC3x-AA-HLCs), a switch to a mature cellular metabolism was also seen and an improved CYP expression pattern, even if CYP3A4 expression and function remained lower than in HC3x-AAGly-HLCs [22].

Here, we demonstrate that metabolically improved HC3x-AA-HLCs can (i) efficiently be infected with HBV and (ii) are able to produce novel HBV virions that (iii) allow secondary rounds of infection, suggesting that HC3x-AA-HLCs support the full HBV life cycle. Additionally, the model was validated with the nucleoside reverse transcriptase inhibitor (-)-beta-L-2′,3′-dideoxy-3′-thiacytidine (3TC), known as lamivudine (LAM) and the HBV entry inhibitor, myrcludex B [4,9].

## 2. Materials and Methods

### 2.1. Cell Cultures

The human embryonic stem cell (hESC; WA09) line H9 and the human induced pluripotent stem cell line (hiPSC-Sigma0028) were purchased from WiCell Research Institute (Madison, WI, US) and Sigma-Aldrich (Saint Louis, MO, USA), respectively. Both cell lines were maintained in feeder-free cultures on human matrigel coated plates (BD Biosciences, San Jose, CA, USA) in E8 Flex culture medium (Gibco) in a humidified 5% CO_2_ incubator at 37 °C. Both hPSC lines were genetically modified to incorporate doxycycline-inducible cassette encoding for *HNF1α*, *PROX1* and *FOXA3*, in the *adeno-associated virus integration site*
*(AAVS1)* locus (termed hESC-HC3x and hiPSC-HC3x), as previously published by Ordovas et al. and Boon et al. [21,22]. The HC3x hPSCs were differentiated as previously described, with minor adjustments [22]. Briefly, HC3x hPSCs were plated on growth-factor-reduced matrigel-coated plates at ±8.75 × 10^4^ cells/cm^2^ in mTeSR1 medium (Stem Cell Technologies, Vancouver, CA, USA). Two days after plating the HC3x hPSCs, the cells were differentiated using a previously described cytokine cocktail [24]. Dimethyl sulfoxide (DMSO) at a concentration of 0.6% was added from day 0 until day 12 and increased to 2% from day 12 of differentiation onwards. In addition, from day 12 of differentiation, liver differentiation medium (LDM) was supplemented with a mix of MEM Non-Essential Amino Acids Solution (100×, 16 mL/100 mL of LDM) and Amino Acids Solution (50×, 8 mL/100 mL of LDM) (AA medium). To induce the overexpression of the three TFs, 5 µg/mL doxycycline (Sigma-Aldrich) was added to the cell culture medium from day 4 onwards. The human hepatoma cell line HepG2, constitutively overexpressing *NTCP* (HepG2-NTCP) (kindly provided by the lab of Prof. Stephan Urban, Heidelberg, Germany), was cultured in Dulbecco’s modified Eagle’s medium (DMEM) (Gibco) containing 10% fetal bovine serum (FBS) (Gibco), 1% non-essential amino acids (NEAA) (Gibco), 1% penicillin-streptomycin (Gibco), 1% L-glutamine (Gibco) and 5 µg/mL Puromycin (Sigma-Aldrich), in a humidified 5% CO_2_ incubator at 37 °C.

### 2.2. Primary Human Hepatocytes

Primary human hepatocytes (PHHs) were kindly provided by Prof. Etienne Sokal (Gastroentérologie Hépatologie Pédiatrique, Cliniques Universitaires St. Luc, Brussels, Belgium). PHHs of three different donors—F110 (Female, 26 years old), F125 (male, 62 years old) and F133 (male, 57 years old)—were collected post-mortem in accordance with the Belgian legislation of tissue donation. The PHHs were used as control for mRNA expression assessment.

### 2.3. Viral Inoculation

Cell culture supernatant of HepG2_AD38 cells (kindly provided by Prof. Stephan Urban, Heidelberg, Germany) with a stably integrated 1.3mer copy of the HBV ayw strain (GenBank accession no. V01460) were collected, and the HBV particles were concentrated via precipitation with PEG-it (System Biosciences) following the manufacturer’s instructions. Subsequently, the precipitated HBV particles were resuspended in PBS containing 20% (vol/vol) glycerol and stored at −80 °C. HC3x-AA-HLCs were infected on day 16 of differentiation with HBV at the indicated multiplicity of infection (MOI). The viral inoculum was supplemented with 2.5% DMSO and 4% polyethylene glycol (PEG). The cells were incubated with the inoculum for 24 h in a humidified 5% CO_2_ incubator at 37 °C. Medium was changed every other day until the end of the experiment.

### 2.4. RNA Extraction and RT-qPCR

RNA lysates were collected and isolated using the GenElute Mammalian Total RNA Miniprep Kit (Sigma-Aldrich). cDNA synthesis was performed using the Superscript III First-Strand synthesis kit (Invitrogen) and followed by qPCR using the Platinum SYBR green qPCR supermix-UDG kit (Invitrogen). Detection was done with the ViiA7 Real-Time PCR instrument (Thermo Fischer Scientific, Waltham, MA, USA). Gene expression was normalized for the expression of housekeeping gene *RPL19*. Sequences of primers are listed in Appendix A.

### 2.5. Myrcludex B Staining

HC3x-AA-HLCs were cultured on cover slips. To functionally define the presence of *NTCP* receptors on the cell membrane, cells were incubated for 30 min at 37 °C with 400 nM Atto^488^-labelled myrcludex B or with an Atto^488^-labelled mutated form of myrcludex B (control), which is unable to bind *NTCP* (kindly provided by the lab of prof. Stephan Urban, Heidelberg, Germany). After incubation, the cells were washed thoroughly with PBS and fixed with 4% paraformaldehyde (PFA) at room temperature. Subsequently, the cells were again washed with PBS and stained with Hoechst (Sigma-Aldrich). The cover slips were mounted, and images were taken with an AxioimagerZ.1 fluorescence microscope (Carl Zeiss Inc., Oberkochen, Germany).

### 2.6. Immunofluorescence Staining

Hepatocyte markers: HC3x-AA-HLCs were fixed with 4% PFA for 15 min at room temperature and washed with PBS. Next, the HC3x-AA-HLCs were permeabilized with 0.2% Triton-X100 in PBS for 15 min at room temperature and blocked with 5% donkey serum in 0.2% Triton-X100 in PBS for 30 min at room temperature. The cells were incubated overnight at 4 °C with anti-HNF4α and anti-NTCP antibodies, followed by appropriate secondary antibodies and Hoechst (Sigma-Aldrich). Antibodies and the dilutions used are listed in Appendix A. Images were taken with the AxioimagerZ.1 fluorescence microscope (Carl Zeiss Inc.).

HBV core and HBV surface antigens: the HBV-infected HC3x-AA-HLCs and HepG2-NTCP cells were fixed with 4% PFA for 15 min at room temperature and washed with PBS. The cells were blocked and permeabilized with a blocking/permeabilization solution (2% fetal calf serum (FCS), 3% bovine serum albumin (BSA) and 0.2% Tween-20 in PBS), for 24 h at 4 °C. After incubation, the cells were incubated for 90 min at room temperature with rabbit anti-HBV core antigen (HBcAg) and mouse anti-HBV surface antigen (HBsAg) in blocking/permeabilization solution. The cells were washed three times for 15 min with 0.05% Triton-X100 in PBS and incubated for 75 min at room temperature with the appropriate secondary antibodies and Hoechst (Sigma-Aldrich) in blocking/permeabilization solution. Fluorescence was measured with a Cell Insight CX5 High Content Screening platform (Thermo Fischer Scientific). Antibodies and the dilutions used are listed in Appendix A.

### 2.7. HBsAg ELISA

For the HBsAg ELISA, the DiaSource kit (KAPG4SGE3) was used following the manufacturer’s protocol. Briefly, 50 µL of Anti-HBs peroxidase solution sample was added to 50 µL sample. After incubation of 90 min in 37 °C, the samples washed thoroughly and incubated for 30 min at room temperature with 50 µL of chromogenic 3,3′,5,5′-Tetramethylbenzidine (TMB) concentrate and 50 µL of substrate buffer. The reaction was stopped with 100 µL stop solution (sulfuric acid), and the absorbance was measured at 450 nm. 620 nm was used as a reference wavelength. Data were normalized by the following formula: (absorbance at 450 nm)/(absorbance at 620 nm).

### 2.8. HBeAg ELISA

For the HBeAg ELISA, the DiaSource kit (KAPG4BNE3) was used following the manufacturer’s protocol. Briefly, 50 µL of neutralizing solution sample was added to 50 µL sample. Samples were incubated for 60 min in 37 °C and washed thoroughly after incubation. 100 µL of anti-HBe-peroxidase was added, and after incubation for 60 min at 37 °C, samples were washed and incubated for 30 min at room temperature with 50 µL of chromogenic 3,3′,5,5′-Tetramethylbenzidine (TMB) concentrate and 50 µL of substrate buffer. The reaction was stopped with 100 µL stop solution (sulfuric acid), and the absorbance was measured at 450 nm. 620 nm was used as a reference wavelength. Data were normalized by the following formula: (absorbance at 450 nm)/(absorbance at 620 nm).

### 2.9. Inhibition Experiments

(-)-beta-L-2′,3′-dideoxy-3′-thiacytidine (3TC; also known as lamivudine (LAM)) was purchased from MedChem Express (cat.no. HY-B0250). Myrcludex B (myrB) was kindly gifted by Prof. Stephan Urban (Heidelberg, Germany). HBV-infected HC3x-AA-HLCs were treated with 0.5 µM LAM or 100 nM myrB, which was added to the medium from viral inoculation until the end of the infection experiments.

### 2.10. Titrations on HepG2-NTCP Cells

Infectious units in the supernatant of HBV-infected HC3x-AA-HLCs, with or without 0.5 µM LAM treatment, were quantified by serial dilution on HepG2-NTCP cells. Following staining for HBc antigen expression and high content imaging analysis, viral titers were calculated by selecting the dilution factor with a fluorescence positive proportion between 10–30% HBc positive cells (linear range) and were expressed as Infectious Units per milliliter (IU/mL).

### 2.11. Statistical Analysis

Data are shown as mean ± SEM. Comparisons between groups were done using Student’s *t*-test and one-way ANOVA, when appropriate. *p*-values < 0.05 (*), *p* < 0.01 (**) and *p* < 0.001 (***), *p* < 0.0001 (****) were considered statistically significant. Analyses were carried out using GraphPad Prism 9.0 (GraphPad Prism Software Inc.).

## 3. Results

### 3.1. HC3x-AA-HLCs Express the HBV Entry Receptor, NTCP

We recently published an improved differentiation protocol, based on the overexpression of 3 TFs (*HNF1α*, *PROX1* and *FOXA3*) and the addition of amino acids and an extra 2% glycine to the culture medium (HC3x-AAGly-HLCs) [22]. Here, we adapted this published protocol for the purpose of in vitro HBV infection studies as the combination of 2.5% DMSO, required for high *NTCP* expression and HBV infection [25,26], and 2% glycine, present in the optimized medium as described by Boon et al., was toxic due to high osmolarity [22]. Therefore, we used a protocol that was based on the addition of both essential and non-essential amino acids but without the addition of 2% glycine. Both hiPSC and hESC HC3x-hPSCs were differentiated towards HLCs using the protocol depicted in Figure 1A. At the end of the hepatocyte differentiation, hESC and hiPSC HC3x-AA-HLCs expressed increased levels of the TFs *HNF1α*, *PROX1* and *FOXA3*, to similar levels observed in PHHs, indicating that the overexpression cassette was functional upon doxycycline induction from day 4 onwards (Figure 1B). In addition, we assessed the levels of hepatocyte marker genes, such as *ALB*, *AAT*, and *HNF4α*, as well as the levels of *NTCP*, which is described as the entry receptor for HBV into the hepatocyte [9], during the differentiation process of hESC and hiPSC HC3x towards HLCs. A continuous increase of the expression of hepatocyte markers *ALB*, *AAT*, *HNF4α* and *NTCP* was observed, and at d22 of differentiation their expression levels were similar to those observed in PHHs (Figure 1C). This was confirmed by immunofluorescence staining of hESC and hiPSC HC3x-AA-HLCs, revealing that nearly all HLCs were HNF4α- and NTCP-positive on day 22 of differentiation and demonstrating that the adapted hepatocyte differentiation protocol generated a highly homogeneous hepatocyte progeny with more than 90% HNF4α^+^ hESC and hiPSC HC3x-AA-HLCs (Figure 1D and Appendix A).

### 3.2. HC3x-AA-HLCs Support HBV Replication and the Production of Novel Infectious HBV Virions

To further demonstrate the presence of the HBV entry receptor *NTCP* on hESC and hiPSC HC3x-AA-HLCs, HC3x-AA-HLCs were incubated with Atto^488^-labeled myrcludex B (myrB). MyrB is an HBV-envelope-protein-derived lipopeptide that can bind *NTCP* [9]. Labelling with Atto^488^-MyrB, but not with Atto488-MyrB mutant, which is unable to bind NTCP, demonstrated the presence of *NTCP* on hESC and hiPSC HC3x-AA-HLCs (Figure 2A). Next, we infected hESC and hiPSC HC3x-AA-HLCs with HBV on day 16 of differentiation (Figure 2B), a time point at which *NTCP* was stably expressed at a high level (Figure 1C). Seven days after infection with the three different inocula (MOI of 0.1, 0.05 and 0.025), the efficiency of infection was determined by several complementary methods. MOI-dependent infection efficiency was observed as shown by immunofluorescence staining for HBcAg and HBsAg in both hESC and hiPSC HC3x-AA-HLCs (Figure 2B–D). However, some variability was observed in infection efficiency between different batches of HC3x-AA-HLCs. Interestingly, the lower infection efficiency was often linked with a higher cell number and lower *NTCP* expression levels (Appendix A). Similarly, gradual increase of released HBsAg (Figure 2E), HBeAg (Figure 2F) and infectious HBV particles (Figure 2G) was shown in the supernatant of the hESC HC3x-AA-HLCs over time. Altogether, these data strongly suggest that HC3x-AA-HLCs are permissive of HBV infection and support replication and the formation of cccDNA (Figure 2E,F).

### 3.3. Validation of HC3x-AA-HLCs with Known Anti-HBV Antivirals

To further validate hESC HC3x-AA-HLCs as a suitable in vitro model allowing screening of HBV antiviral compounds, both myrcludex B (myrB), an HBV entry inhibitor, and 3TC ((-)-beta-L-2′,3′-dideoxy-3′-thiacytidine, lamivudine (LAM)), a known anti-HBV NRTI, were used [4]. MyrB (100 nM) was added to the hESC HC3x-AA-HLC culture prior to HBV infection, resulting in significant albeit incomplete inhibition of HBV infection, as was demonstrated by a reduction in the number of HBc-positive cells (Figure 3A). In parallel, hESC HC3x-AA-HLC culture was treated with LAM (0.5 µM) upon HBV infection. At the indicated days post infection, supernatant of HBV-infected hESC HC3x-AA-HLCs, treated with or without LAM, was collected and titrated on HepG2-NTCP cells for the presence of infectious virus particles (Figure 3B). As described above (Figure 2G), an increase of secreted HBV infectious progeny over time was observed that could be completely inhibited by LAM treatment. In addition, we assessed whether hESC HC3x-AA-HLCs could also support viral spreading by monitoring the number of HBcAg-positive cells and release of HBsAg in HBV-infected HC3x-AA-HLCs treated with or without LAM during a period of 14 days (Figure 3C,D). In the absence of LAM, a gradual increase of HBc-positive cells and released HBsAg could be observed over time, as previously described (Figure 2C–F). In the presence of LAM, however, no clear increase of HBc-positive cells and released HBsAg could be observed. Furthermore, there was no clear difference in the LAM-treated cells between day 4 and day 13 post infection on the number of HBc-positive cells and the amount of released HBsAg.

Together, our data suggest an ongoing virus replication with an increased viral spreading over time that can be prevented by LAM treatment and further underscores the potential of hESC HC3x-AA-HLCs to support the HBV replication cycle and the potential to be used as an in vitro tool to screen novel anti-HBV antiviral compounds.

## 4. Discussion

Worldwide, about 300 million people are chronically infected with HBV. Although NRTIs are highly efficient, life-long treatment is necessary as the cccDNA is retained in the hepatocyte nucleus [1,2,3]. Hence, there is an urgent need for therapeutics that can cure chronic HBV patients rather than control the disease. PHHs are currently the gold standard for in vitro HBV studies [25]. However, they have their drawbacks; thus, alternatives for HBV studies are needed [27]. Therefore, the use of hPSC-HLCs as in vitro HBV infection models has been investigated recently [7,13]. Nevertheless, even the most ‘mature’ hepatocyte progeny derived from hPSCs still resembles fetal hepatocytes, as has been observed by transcriptomics, proteomics and functional studies, and this irrespective of the differentiation process researchers have used [18,19,20]. We recently created hPSC lines overexpressing three transcription factors (HC3x hPSCs; *HNF1α*, *PROX1* and *FOXA3*) that, when cultured in metabolically optimized medium conditions (containing high levels of a mixture of amino acids (AA)), generate a more mature hepatocyte progeny with increased drug metabolizing capacity, as well as improved cellular metabolism, i.e., oxidative phosphorylation for energy metabolism and gluconeogenesis, rather than glycolysis, and amino acid dose-dependent improved drug biotransformation capacity [22]. Here, we tested if these metabolically improved HC3x-AA-HLCs could be used as an in vitro infection model for HBV.

We demonstrated that HC3x-AA-HLCs express high levels of the HBV entry receptor, NTCP, from day 16 of differentiation onwards and that Atto^488^-labbeled myrB, an HBV-envelope-protein-derived lipopeptide, could bind the *NTCP* expressed on the cell surface of HC3x-AA-HLCs [9]. Consistently, HC3x-AA-HLCs could efficiently and in a dose-dependent manner be infected with HBV. This was shown by immunofluorescence staining for HBcAg and HBsAg. Furthermore, HBsAg and HBeAg were secreted in the HLC supernatant, which increased over time, suggesting that functional cccDNA is formed in the HBV-infected HC3x-AA-HLCs. In line with this observation, re-infection and back-titration studies on HepG2-NTCP cells showed that the HBV-infected HC3x-AA-HLCs produced novel infectious HBV virions, suggesting the formation of cccDNA. However, quantification of cccDNA via qPCR or southern blotting has to be performed to further confirm the presence and amplification of cccDNA in HBV-infected HC3x-AA-HLCs. Treatment of HBV-infected HC3x-AA-HLCs with lamivudine blocked the production of novel HBV viral particles, indicating that the infectious particles detected in the supernatant of untreated HBV-infected HC3x-AA-HLCs are newly produced HBV infectious particles, not originating from the initial HBV inoculum. Based on the increase in the percentage of HBc^+^ cells and the levels of HBs in the supernatant of untreated HBV-infected HC3x-AA-HLCs, these results suggest that this in vitro model supports HBV viral spreading over time, a feature which has not been observed in HBV-infected HepG2-NTCP cells [10]. Thus, HC3x-AA-HLCs can be used as an in vitro tool to study the HBV life cycle and viral spreading.

hPSC-HLCs were described to support HBV viral infection, but their infection efficiency was very low, even with the addition of a Janus kinase inhibitor (JAKi) to block the innate immune system [7]. By contrast, we found here that even without manipulating the innate immune response, HC3x-AA-HLCs were infected following exposure to a relatively low MOI of HBV, compared to previously published hPSC-HLC HBV infection studies [7,13]. As result, the more mature HC3x-AA-HLCs appear to be highly permissive for HBV, constituting an interesting tool to study HBV in vitro and to use this model to screen potential anti-HBV antivirals [7,13]. Nevertheless, we observed some variability in the infection efficiency of the HC3x-AA-HLCs. A possible explanation for the variability in infection efficiency of the HC3x-AA-HLCs might be density of the HC3x-AA-HLCs at the time of infection, which appeared to be linked to the expression levels of the HBV entry receptor, *NTCP*. Accordingly, HBV infection efficiency was inversely correlated with higher cell density and lower *NTCP* expression levels on day 16 of differentiation. These lower *NTCP* expression levels may underlie the decrease seen in the percentage of HBV-infected cells as protein levels of *NTCP* and HBV infection are strongly correlated [28]. However, more studies are needed to confirm this observation and to further optimize the HC3x-AA-HLC cell density during hepatocyte differentiation to obtain a more consistent HBV infection efficiency between batches of differentiations. Overcoming this batch-to-batch variability will be of great importance to enable medium and high-throughput screens for novel candidate anti-HBV antivirals and to study in vitro HBV infection and replication in general.

Importantly, the HC3X-AA-HLC model could correctly identify the antiviral effect of the HBV entry inhibitor, myrcludex B, and nucleoside analogue, lamivudine [4,9,29,30]. A significant reduction of HBV infection in HC3x-AA-HLCs was observed upon myrcludex B treatment. Lamivudine treatment completely inhibited the release of infectious HBV particles and HBsAg and prevented viral spreading over time. As with all other anti-HBV antivirals on the market, lamivudine only inhibits HBV replication while not eliminating HBV cccDNA. The reasons for failure to identify drugs that can eliminate cccDNA using the current models include (1) the poor stability of PHHs cultured in classical 2D cultures with loss of hepatic function; and (2) the inability of HepG2-NTCP cells to recapitulate the full HBV replication cycle as no viral spread is observed upon HBV infection. However, we previously demonstrated that HC3x-AA-HLCs can be maintained in culture for at least 50 days without losing their hepatocyte phenotype and that this culture system can be downscaled to 384-well format culture plates, which should enable large-scale small molecule screens to identify novel anti-HBV drugs to improve and expand the existing therapeutic regimens [22].

Finally, resistance of HBV has been extensively reported for lamivudine [31,32], and it has been shown that this is associated with acquisition of drug-resistant mutations in the RNA-dependent DNA polymerase of HBV. As the HC3x-AA-HLC model is known to be stable at least until day 50 of differentiation, it would be of great interest to explore if the appearance of resistant virus variants to drugs such as lamivudine can also be detected following long-term culture with suboptimal drug concentrations.

To conclude, here, we demonstrate that the HC3x-AA-HLCs can efficiently be infected with HBV, as was shown by the expression of HBcAg and HBsAg. A clear and increasing release of HBsAg and HBeAg was detected, indicating the formation of functional cccDNA. Moreover, back-titration of the culture supernatant of HBV-infected HC3x-AA-HLCs on HepG2-NTCP cells revealed the production of novel HBV particles. An increasing number of HBcAg-positive HC3x-AA-HLCs over time suggests that viral spreading is occurring. Finally, the HC3x-AA-HLC model was validated for its use in antiviral drug studies.

## Figures and Tables

**Figure 1 biomedicines-10-00268-f001:**
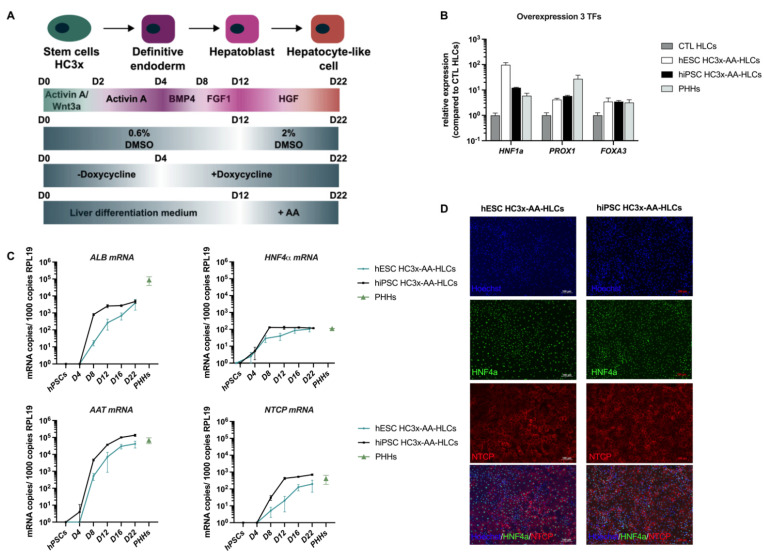
HC3x hPSCs differentiate towards HLCs and express high levels of hepatocyte markers and HBV entry receptor NTCP. (**A**) Optimized differentiation protocol of HC3x hPSCs towards HLCs for the purpose of in vitro HBV infection studies. (**B**) Gene expression for transcription factors *HNF1α*, *PROX1* and *FOXA3* in CTL HLCs (no doxycycline-induced transcription factor overexpression), hESC and hiPSC HC3x-AA-HLCs on day 22 of differentiation and in PHHs. (**C**) Gene expression for the hepatocyte markers *ALB*, *AAT*, *HNF4α* and *NTCP* in hESC and hiPSC HC3x-AA-HLCs during the hepatocyte differentiation, in hPSC stage and in PHHs. (**D**) Immunofluorescence staining of HNF4α and *NTCP* in hESC and hiPSC HC3x-AA-HLCs on day 22 of differentiation. These images are representative of three independent experiments (Scale bar = 100 μM). All data are represented as mean ± SEM.

**Figure 2 biomedicines-10-00268-f002:**
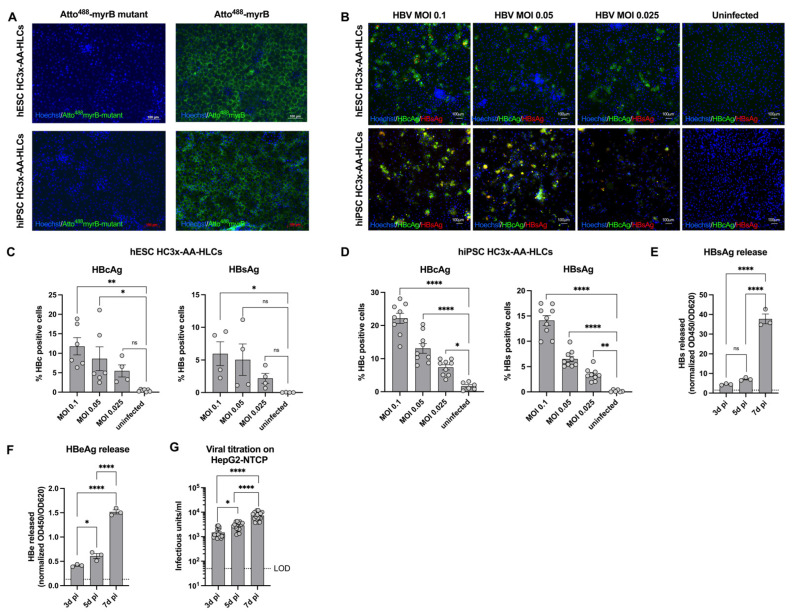
HC3x-AA-HLCs are susceptible to HBV infection and are able to support HBV replication and the production of novel infectious HBV virions. (**A**) hESC and hiPSC HC3x-AA-HLCs were stained with Atto^488^-myrB at the end of differentiation. A mutant Atto^488^-myrB was used as a negative control. These images are representative of three independent experiments (Scale bar = 100 μM). (**B**) hESC and hiPSC HC3x-AA-HLCs were infected at day 16 of differentiation with HBV at a MOI of 0.1, 0.05 or 0.025. At 7 d pi, the cells were fixed and immunofluorescence staining for HBcAg and HBsAg was performed. Uninfected HC3x-AA-HLCs were used as negative control. These images are representative of at least three independent experiments (Scale bar = 100μM). (**C**) Quantification of the percentage HBcAg^+^ and HBsAg^+^ HBV-infected hESC-HC3x-AA-HLCs. (**D**) Quantification of the percentage HBcAg^+^ and HBsAg^+^ HBV-infected hiPSC-HC3x-AA-HLCs. (**E**) ELISA for HBsAg in the supernatant of uninfected and HBV-infected HC3x-AA-HLCs (HBV MOI 0.1) on 3, 5 and 7 d pi (N = 3). (**F**) ELISA for HBeAg in the supernatant of uninfected and HBV-infected HC3x-AA-HLCs (HBV MOI 0.1) on 3, 5 and 7 d pi (N = 3). (**G**) Quantification of the viral titration of 3, 5 and 7 d pi supernatant of HBV-infected HC3x-AA-HLCs (HBV MOI 0.1) on HepG2-NTCP cells (N = 3). All data are represented as mean ± SEM. *p*-values < 0.05 (*), *p* < 0.01 (**) and *p* < 0.0001 (****).

**Figure 3 biomedicines-10-00268-f003:**
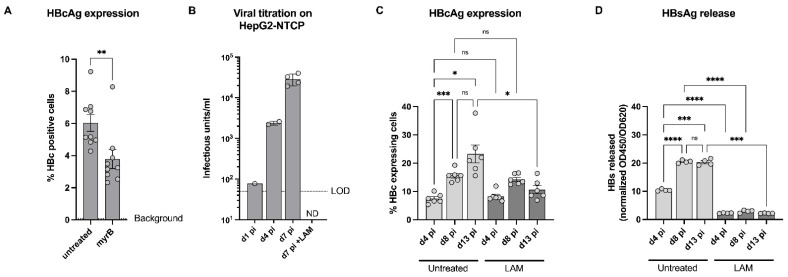
Validation of HC3x-AA-HLCs with known anti-HBV antiviral, lamivudine. (**A**) Quantification of the percentage HBcAg^+^ HBV-infected HC3x-AA-HLCs (HBV MOI 0.1), with or without 100 nM myrcludex B (myrB) treatment on 7 d pi. (**B**) Quantification of the viral titration of 1, 4 and 7 d pi supernatant of HBV-infected HC3x-AA-HLCs (HBV MOI 0.1), with or without 0.5μM lamivudine (LAM) treatment, on HepG2-NTCP cells. (**C**) Quantification of the percentage HBcAg^+^ HBV-infected HC3x-AA-HLCs (HBV MOI 0.1), with or without 0.5 μM lamivudine (LAM) treatment, on 4, 8 and 13 d pi. (**D**) ELISA for HBsAg in the supernatant of HBV-infected HC3x-AA-HLCs (HBV MOI 0.1), with or without 0.5μM lamivudine (LAM) treatment, on 4, 8 and 13 d pi. All data are represented as mean ± SEM. *p*-values < 0.05 (*), *p* < 0.01 (**) and *p* < 0.001 (***), *p* < 0.0001 (****).

## Data Availability

Not applicable.

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
