# Peer review of "Metabolically Improved Stem Cell Derived Hepatocyte-Like Cells Support HBV Life Cycle and Are a Promising Tool for HBV Studies and Antiviral Drug Screenings"

_biomedicines, 2022, doi:10.3390/biomedicines10020268_

Round 1

Reviewer 1 Report

The authors report on  the use of hPSC-derived HLCs for in vitro HBV studies.
The topic is interesting and relevant for the journal. Indeed, due to limitations in using PHHs, this can be an useful, alternative approach. The manuscript is well organized and data support author conclusions in terms of expression of relevant markers, immunosfluorescence staining, HBV replication, novel infectious virions, antiviral drug studies. I suggest publication in the present form.

Author Response

We would like to thank the reviewer for reviewing the manuscript and we appreciate the positive comments.

Reviewer 2 Report

The authors demonstrated that the human pluripotent stem cell-hepatocyte-like cells generated using their protocol, could support efficient hepatitis B virus infection and have the potential to be used as an in vitro HB infection model for anti-viral drug studies. Please see below for some comments:

  1. In figure 1, only the mRNA expression of HNF1a, PROX1, and FOXA3 were presented. What about the protein level expression of those genes or the protein expression of hepatocyte markers?
  2. What is the maturity of those hPSC-HLCs when compared to other published hPSC-derived hepatocytes and primary human hepatocytes?
  3. What is the purity of hPSC-HLCs used in those experiments and will the purity affect the results of the study?
  4. In Figure 1C, what is the expression level of those maturation markers in primary human hepatocyte and hPSCs used for HLC differentiation?
  5. The quality of immunofluorescence staining images in the manuscript needs to be improved. The images are blurry and it’s hard to see positive staining in merged images. High magnification images or enlarged positive staining areas are needed.
  6. The scale bars are missing in Figure 2B.
  7. Only lamivudine was tested in this study. Why not other anti-HBV antiviral compounds such as tenofovir alafenamide, entecavir, and telbivudine? Will HC3x-AA-HLCs have a similar effect?

Author Response

We would like to thank the reviewer for the constructive feedback. Please find our answers in the attachment.

Round 2

Reviewer 2 Report

The authors addressed my concerns.